# Evaluating Pedometer Algorithms on Semi-Regular and Unstructured Gaits [note 1]

**DOI:** 10.3390/s21134260

**Published:** 2021-06-22

**Authors:** Ryan Mattfeld, Elliot Jesch, Adam Hoover

**Affiliations:** 1Computer Science Department, Elon University, Elon, NC 27244, USA; 2Department of Food, Nutrition, and Packaging Sciences, Clemson University, Clemson, SC 29634, USA; ejesch@clemson.edu; 3Holcombe Department of Electrical and Computer Engineering, Clemson University, Clemson, SC 29634, USA; ahoover@clemson.edu

**Keywords:** accelerometer dataset, mHealth, multiple gaits, pedometer, wearable sensors

## Abstract

Pedometers are popular for counting steps as a daily measure of physical activity, however, errors as high as 96% have been reported in previous work. Many reasons for pedometer error have been studied, including walking speed, sensor position on the body and pedometer algorithm, demonstrating some differences in error. However, we hypothesize that the largest source of error may be due to differences in the regularity of gait during different activities. During some activities, gait tends to be regular and the repetitiveness of individual steps makes them easy to identify in an accelerometer signal. During other activities of everyday life, gait is frequently semi-regular or unstructured, which we hypothesize makes it difficult to identify and count individual steps. In this work, we test this hypothesis by evaluating the three most common types of pedometer algorithm on a new data set that varies the regularity of gait. A total of 30 participants were video recorded performing three different activities: walking a path (regular gait), conducting a within-building activity (semi-regular gait), and conducting a within-room activity (unstructured gait). Participants were instrumented with accelerometers on the wrist, hip and ankle. Collectively, 60,805 steps were manually annotated for ground truth using synchronized video. The main contribution of this paper is to evaluate pedometer algorithms when the consistency of gait changes to simulate everyday life activities other than exercise. In our study, we found that semi-regular and unstructured gaits resulted in 5–466% error. This demonstrates the need to evaluate pedometer algorithms on activities that vary the regularity of gait. Our dataset is publicly available with links provided in the introduction and Data Availability Statement.

## 1. Introduction

Fitness trackers motivate increased physical activity by counting steps during everyday life [1]. In 2016, 10% of the U.S. population owned at least one wearable fitness device [2]. Fitness and lifestyle wearable products are expected to experience huge growth as advances in research make better use of sensors [3]. Fitness tracker sales have increased since 2016 and are projected to continue increasing [4]. Recommendations for daily step count range from 8000 to 12,000 steps depending on age and gender [5], with the goal of 10,000 steps/day commonly used to motivate increased physical activity [6]. Pedometer accuracy is important because inaccuracy can lead to user frustration and low compliance with device usage [7]. Studies have found that pedometer accuracy is generally high during exercise activities [8]. However, in the U.S. people dedicate about 30 min per day on average to exercise activities [9]. This means that a substantial number of steps are being taken and counted during activities other than exercise. Several studies conducted in free-living conditions have found large differences in pedometer accuracy by comparing steps counted by different pedometer devices worn at the same time [10,11,12,13].

The main contribution of this paper is that we are the first to look at the effect of regularity of gait on pedometer accuracy. All previous works evaluated accuracy while participants walked with a regular gait resembling mild exercise. Specifically, subjects were asked to walk for a period of time or distance or step count, with no breaks or other intermittent activities. We hypothesize that disruptions in walking throughout everyday life are a large contributor to pedometer inaccuracy. We propose that *consistency of gait* is a critical variable to analyze when evaluating accuracy, and we define and test three levels of this variable: regular gait; semi-regular gait; and unstructured gait. We define regular gait as what occurs during exercise and other daily activities over extended periods (5+ min) of consistent walking that is uninterrupted. We define semi-regular gait as what occurs when moving through buildings, which is primarily composed of periods of time that resemble regular gait broken up by brief interruptions (e.g., stopping, starting, or changing direction). We define unstructured gait as what occurs when performing activities within a room, which is primarily composed of very brief periods of regular gait (approximately 3–10 steps) with more frequent interruptions including periods of rest and change of direction. Because step detection algorithms were originally designed for exercise, where many similar steps are taken in a consistent pattern, we hypothesize that these algorithms perform poorly on the more interrupted, inconsistent motion patterns in everyday life that occur with semi-regular and unstructured gait. Evaluating step detection algorithms on these gaits will result in pedometers that are more accurate during everyday use, meeting modern demands.

To test our hypothesis, we collected a new data set in which gait consistency was manipulated by having subjects perform three different tasks resembling common everyday life activities [14]. We evaluated the three most common types of pedometer algorithms including one based on peak detection [15], one based on threshold crossing [16], and one based on autocorrelation [17]. The accuracy of these algorithms on regular gait data, regardless of body position, were all high (3–8% error). However, they all decreased significantly when analyzed on semi-regular and unstructured gaits (5–466% error). This demonstrates the need to evaluate pedometer algorithms on activities that vary the regularity of gait. Besides evaluating the effect of regularity in gait, this also allows us to determine whether any particular type of algorithm performs better or worse specifically as gait regularity varies. In addition to our own dataset, we include two other publicly available datasets that recorded regular gait activities [18,19]. We do this to determine whether regular gait accuracy evaluation can be replicated across different datasets. The main contribution of this paper is to evaluate pedometer algorithms when the consistency of gait changes to simulate everyday life activities other than exercise. Finally, we recognize that many results from wearable device analyses are inhomogeneous [20] and, in an effort to provide opportunities for algorithms to be tested on identical datasets, we have made our data publicly available at https://sites.google.com/view/rmattfeld/pedometer-dataset (accessed on 21 June 2021).

This paper is organized as follows: In Section 2, we describe related works and identify regularity of gait as a variable in pedometer algorithm evaluation that has not yet been researched. In Section 3, we describe the data collection process, ground truth step identification process and the process used to evaluate three state-of-the-art pedometer algorithms. Section 4 provides the results of evaluating the three pedometer algorithms on our dataset as well as two other publicly available datasets. Section 5 discusses our findings.

## 2. Related Works

Previous works searching for the cause(s) of variability in pedometer accuracy have tested many variables [21], including the location on the body the device is worn, carried, or used [22,23,24,25,26,27], the device model or step counting algorithm [23,24,28,29,30,31], walking speed [23,24,25,26,28,30,32,33,34], age [32], presence of gait aids [35], weight [32], type of surface [33,34,36], and distance [31]. These variables have all been found to influence pedometer accuracy, as seen in Table 1. In this table, it can also be seen that our work is the first to examine pedometer accuracy as regularity of gait changes. It can also be seen that the regularity of gait affects pedometer accuracy to a larger degree than the previously examined variables.

For example, a test of the accuracy of 10 different popular wearable and smartphone pedometers, worn at three different body positions (waist, wrist, and pocket) and evaluated at a regular walking pace, found errors ranging up to 27% [29]. A test of the effect of distance on five wearables and four smartphone apps was conducted over 100, 500, and 1500 steps at a regular walking pace, finding errors ranging from 5–40% [31]. Walking speed has been varied by evaluating participants on treadmills taking 100 steps at varied speeds, identifying errors ranging from 17–96% [28]. Age, weight and walking speed have been tested by measuring pedometer accuracy in participants with varying ages, weight classifications and walking speeds, finding accuracies ranging from 2–44% [32]. The effect that a gait aid has on pedometer accuracy has also been evaluated, showing that a four-wheeled walker has the most significant effect with an accuracy of 8.7%, while canes, crutches and stationary walkers also affected pedometer accuracy with accuracies ranging from 97.2% to 36.1% [35].

All of these prior works collect data by having participants walk a set number of steps, a set distance, or a set speed on a treadmill. None alter the regularity of gait by allowing for the pauses, transitions, or interruptions that are common in everyday motion. While these changes in the regularity of gait were not important historically, when pedometers were only used during exercise, they are extremely important today, when smartphones, activity monitors and smart watches count steps as a motivator for increased daily physical activity. These devices are worn throughout the entire day, not just during exercise, and the number of steps reported provide insight into the wearer’s activity levels throughout the day. Since many of these steps occur within gaits that include irregular patterns of motion, it is important that pedometer algorithms are developed to perform well on these gait types.

## 3. Methods

The goal of this experiment is to evaluate the effect of a novel source of error, regularity of gait, on standard pedometer algorithms. First, we instrumented participants with accelerometers and had them conduct activities that varied the regularity of gait. Second, we used synchronized video to annotate the accelerometer data denoting the ground truth times of all steps. Third, we searched a publication database to identify a representative set of popular pedometer algorithms. Lastly, we evaluated the accuracy of these pedometer algorithms as the error source is varied. Note that the goal of this experiment was to determine the effect of the error source on pedometer accuracy. It was not our intent to identify the best possible pedometer algorithm. We seek to determine whether the novel error source is something that needs to be considered in future pedometer algorithm designs.

### 3.1. Data Collection

The study was approved by the Clemson University Institutional Review Board for the protection of human subjects (IRB Number: IRB2017-048). Participants were recruited vie email and provided a $20 Amazon gift card for their participation. In all, 30 participants were selected, including 15 females and 15 males with an average height of 67.3±4.3 inches (172±10.9 cm), mass of 155.5±38.8 lbs (70.5±17.6 kg), and age of 22±2.4 years. Each subject provided informed consent and filled out a Physical Activity and Readiness Questionnaire (PAR-Q) [37]. Participants provided height, mass, and gender information. Throughout the data collection process, each participant wore 3 Shimmer3 devices, located on the wrist, hip and ankle, as shown in Figure 1. The Shimmer3 devices recorded at 15 Hz with raw acceleration measurements ranging from −2 to 2 Gravities and a noise density of 125 µg/$\sqrt{Hz}$. They were synchronized to a single computer’s clock. Each participant also wore a Fitbit Charge 2, located directly adjacent to the Shimmer3 device worn on the wrist. The participant’s feet and legs were recorded throughout the experiment as shown in Figure 2.

Each participant was asked to perform three activities, each designed to elicit a gait seen in everyday life. In the regular gait activity, each participant was instructed to walk two laps around an outdoor path approximately 1300 feet (400 m) long at their normal walking pace. The path contained four turns. This pattern is common in exercise, where steps are not interrupted. In the semi-regular gait activity, participants were instructed to perform a scavenger hunt, locating four objects in four different rooms throughout a building. This is representative of walking around a building, including stopping to open or close doors and walking up and down stairs. Within-building activities producing this type of motion include factory work and shopping. In the unstructured gait activity, participants were asked to build a small Lego toy by assembling pieces distributed among 12 small bins around a room. Participants were only allowed to gather one bin of pieces at a time and to construct the toy at a central location. This pattern is common in activities requiring periods of work with no steps interspersed with short periods of walking, in which participants do not reach a steady stride. Within-room activities producing this type of motion include office work and meal preparation.

### 3.2. Ground Truth

The ground truth process was completed with the assistance of a custom tool described in [14]. Notably, the annotation process revealed that steps could be categorized into two distinct groups, which we call steps and shifts. In the regular gait activity, because the motion of walking is repetitive and uninterrupted, each step taken can be clearly counted and evaluated. Virtually all steps in the regular gait activity can be defined by the inclusion of three elements: (1) the foot moves; (2) the body weight shifts in the direction of the foot movement; and (3) the action takes place within a repeating pattern. An activity meeting these criteria is called a step, and the accelerometer signal generally resembles the signal shown in Figure 3. In this figure, which displays acceleration captured from the vertical axis of the ankle accelerometer, it can be seen that between every right step (red solid line) and left step (blue dashed line), there is a clear peak in the acceleration signal as the left leg moves forward.

While steps do occur within the semi-regular and unstructured types of gait, a significant number of motions occur which contain: (1) a foot movement; (2) may or may not include a weight shift; and (3) are not within a repeating pattern. We specifically refer to these types of motions as a shifts, and these shifts are most commonly identified as (a) a step which causes a sharp change in direction or pace or (b) a step which begins movement from a complete stop or ends movement. Examples of accelerometer signals produced when shifts occur are displayed in Figure 4. The acceleration signals shown in the figure are taken from the vertical axis of the ankle accelerometer. The first two rows in the figure demonstrate the first and last steps taken and produce a smaller magnitude of acceleration compared to steps taken while in stride. The third row demonstrates a pivot step, which occurs as the participant changes direction. The fourth row demonstrates a shuffle, in which the participant moved their foot and shifted their weight during the unstructured gait activity. Steps and shifts were differentiated based solely on video review and were identified separately during the ground truth process.

### 3.3. Inter-Rater Reliability

In order to investigate inter-rater reliability, each of 3 raters labeled all the steps for 3 participants across all 3 types of gait, resulting in an average of 6187 steps and shifts labeled per reviewer. The reviewers varied an average of 0.2% in terms of total steps labeled. This is a strong but not perfect level of agreement due to differences in vigilance in the labeling task and potentially differences in opinions on what is a shift (is there enough movement in the foot to be considered a shift). The average percentage difference in the number of shifts labeled was 0.1% in regular gait, 3.5% in semi-regular gait, and 3.3% in unstructured gait. These differences result from the difficulty of accurately differentiating between steps and shifts. These differences often relate to the “within a repeating pattern” and “sharp change in direction” portions of the shift and step definitions provided. Taken together, this process indicates that vigilance can affect the labeling process, but the difficulty in differentiating steps from shifts or shifts from non-steps is the major factor in labeling differences.

### 3.4. Pedometer Algorithms

In order to identify algorithms for implementation and testing, we searched for papers describing pedometer algorithms. We searched IEEE Xplore using the search terms “pedometer” and “algorithm”. In addition to the 60 papers yielded by this search, related works were found through bibliography searches, and 84 papers were reviewed. Of these, 24 implemented pedometer algorithms and provided enough detail to implement the step detector. From this subset of papers, algorithms were categorized into three groups based on the methodology used in step identification: peak detection, threshold crossing and autocorrelation. These results are summarized in Table 2. It is important to note that there are differences between step detection/counting and other types of analysis, which may use detected steps as input such as gait analysis, activity recognition, and distance tracking [38]. Our literature search focused on the former. The methods of each algorithm are each briefly summarized in the following sections. When evaluating each algorithm, the parameters used were normalized (e.g., if originally trained on accelerations using m/s^2^, parameters were divided by 9.8 to convert to gravities on our dataset) and interpolated (e.g., if time between steps was between 100 and 200 indices when sampled at 100 Hz, this would become 15 and 30 indices when sampled at 15 Hz) as needed. The parameters were then varied across a range of values, as shown in each algorithm’s flow chart. Whichever combination of parameters yielded the highest accuracy for the regular gait activity using the wrist-worn sensor was used throughout further testing in our dataset. The wrist-worn sensor was selected for training because most commercial pedometers are wrist-worn, as they are integrated into smart watches or designed to be worn as a watch.

#### 3.4.1. Peak Detector

We implemented the algorithm proposed by Gu et al. to be representative of peak detection algorithms because it was recently published (2017) and provides enough detail for reimplementation [15]. The algorithm is summarized in Figure 5. The algorithm first finds the magnitude of acceleration calculated as the square root of the sum-of-squares of the X, Y, and Z accelerations. Once this is calculated, peaks in the signal are identified. Next, the motion state is calculated as idle, walking, or running based on the variance in the accelerometer signal between peaks (note we used one threshold to classify data as either idle or walking because running was not present in our dataset). Each peak must also meet thresholds on periodicity, similarity and continuity. If all threshold requirements are met, the peak is counted as a step.

#### 3.4.2. Threshold Crossing

We selected the algorithm developed by Zhao to be our representative sample because the algorithm provided sufficient detail for reimplementation and utilized techniques common across many threshold-crossing based algorithms [16]. The algorithm first smooths the accelerometer signal. Then, a dynamic zero crossing threshold is calculated by averaging the maximum and minimum values at 1/2 s intervals. Accelerometer values are required to surpass a precision threshold in order to update stored values. If the change in slope is negative when the acceleration values cross the threshold, a potential step is detected. These steps are summarized in the flowchart shown in Figure 6. Then, a minimum and maximum time requirement between steps is applied, and a requirement that at least four consecutive valid steps must be found in order for step detection to occur as shown in Figure 7. These additional conditions filter out many potential false positives.

#### 3.4.3. Autocorrelation

Of the two algorithms using autocorrelation, we reimplemented the algorithm described by Rai et al. [17]. Both papers using this algorithm are from the same research group, so we used the paper providing a greater level of detail regarding algorithm implementation. The process used by the algorithm is summarized in the flow chart shown in Figure 8. The algorithm calculates the normalized autocorrelation value cross a range of window sizes within a minimum and maximum time range. Whichever window size produces the highest autocorrelation value is used, and the range of window sizes to test on future iterations are updated based on the optimum window size found. Once autocorrelation windows are found, an additional check is performed to detect segments in which participants are idle based on the magnitude of acceleration, and these segments are removed from further consideration. For all segments not labeled as idle, a threshold is applied to the normalized autocorrelation score. If the threshold is surpassed, the data is identified as walking, and the rate of step detection is based on the optimum window size.

### 3.5. Datasets

In addition to testing each of the three algorithms on our dataset, we tested each algorithm on two publicly available datasets: the MAREA (Movement Analysis in Real-world Environments using Accelerometers) Gait Database [18] and the Sensor-based Gait Analysis Validation Data dataset [19]. These datasets were selected by searching for public datasets that identify individual steps taken by participants: these datasets could be used to evaluate the detection of individual steps. Each dataset was developed to test different criteria, as summarized in Table 3. The MAREA dataset includes walking and running, both indoors and outdoors, but only includes regular types of gait. The Kluge dataset includes healthy people and people with Parkinson’s disease, but only includes regular types of gait. Our dataset is the first to include activities with less regular types of gait, allowing for the assessment of pedometers when applied to the irregular motions that are common in real world activity. These types of gait include a greater proportion of shifts, as described above, and detecting shifts in an accelerometer signal can be particularly difficult.

### 3.6. Evaluation

Each of the pedometer algorithms was evaluated by comparing the number of steps detected by the algorithm against the number of steps actually taken according to the ground truth labels for each participant and activity. This is referred to as running count accuracy (RCA), a metric which is commonly used in pedometer evaluation. RCA is a comparison of the number of ground truth steps taken over the course of an activity against the number of steps reported by a pedometer algorithm. The equation used in this work to calculate RCA is shown in Equation (Equation 1).
(1)$$RCA=\frac{\#DetectedSteps}{\#GroundTruthSteps}.$$

An ideal RCA value is 1, with values greater than 1 indicating the algorithm overestimates steps, and values less than 1 indicating the algorithm underestimates steps.

Because individual steps were identified in our dataset, we were also able to use the F1 score as a metric. To calculate this, true positives were identified when a step detected by an algorithm occurred within 0.5 s of a ground truth step. Once a step detected by the algorithm and a ground truth step were matched as a true positive pair, they were excluded from being paired with any additional steps. Once the matching process was complete for an activity, any ground truth steps that were not matched were considered false negatives and any detected steps which were not matched were considered false positives. These values were then used to calculate precision and recall, and from these the F1 score was calculated and reported.

## 4. Results

### 4.1. Algorithm Evaluation

We examined the accuracies for each algorithm across all datasets and across regular, semi-regular, and unstructured gaits. Presenting all potential evaluations is challenging because the dimensionality of the analysis grows quickly when considering three sensor positions, three types of gait, three algorithms evaluated, three datasets used, and (in the Kluge dataset) two health conditions. In addition, in our dataset, shifts could be included or excluded from consideration as ground truth steps, yielding 324 potential accuracy measures. Sensor position and health condition were found to affect accuracy to a lesser degree than the other metrics, so accuracies were averaged across these dimensions.

The Running Count Accuracies (RCA) of each pedometer algorithm across each dataset and gait type can be seen in Table 4. These results demonstrate that there are large differences in RCA across the types of gait observed. In addition, we draw three conclusions from this data. First, all algorithms showed similar accuracy regardless of location worn during regular gait, with a wrist accuracy of 0.98±0.12, hip accuracy of 1.01±0.23, and an ankle accuracy of 0.91±0.19. This demonstrates that many algorithms work well on regular gait regardless of where the sensor is worn, and that all three datasets are capable of performing this evaluation. Second, the peak detector and threshold based algorithms had much worse accuracy for semi-regular and unstructured gaits (1.30 to 5.66) compared to their accuracies on regular gaits (0.92 to 1.03). This demonstrates that semi-regular and unstructured gaits present very different challenges to pedometer algorithms compared to regular gaits. Pedometers using these algorithms significantly overestimate step count during normal daily living. Third, the accuracy of the autocorrelation algorithm did not vary nearly as much across gait type (0.93 to 1.11) as did the accuracies of the peak detector and threshold-based algorithms (0.92 to 3.09 and 1.03 to 5.66, respectively). This demonstrates that some algorithms used for step detection may perform better on the less regular gaits common in everyday life. This is especially important for pedometers that are intended to be worn all day, when gait type can be expected to change repeatedly.

Figure 9 demonstrates how pedometer accuracy decreases on semi-regular and unstructured gait, compared to regular gait. Blue lines indicate ground truth steps. Green lines indicate true positive steps detected by the peak detector algorithm, and red lines indicate false positives. In Figure 9a, it can be seen that, for regular gait, the detected steps are all true positives. In Figure 9b, it can be seen that, during semi-regular gait, as the participant opened a door at the end of a hall, transitioning from walking down a hall to walking on stairs, four false positives were detected. In Figure 9c, it can be seen that while the participant was walking during unstructured gait, the peak detector did well, but it picked up many false positives as the participant constructed their Lego. The false positives were caused by detecting peaks in acceleration when no steps were taken. This example demonstrates that a large number of false positives are detected when the gait is not regular.

Because individual steps are labeled within the data, F1 score was also calculated. When examining F1 scores calculated for the dataset, two interesting results were found. First, when F1 score was examined across sensor position, the wrist sensor yielded an average F1 score of 0.69, the hip 0.73, and the ankle 0.76. Similarly, when examined across gait type, regular gait resulted in an average F1 score of 0.90, semi-regular 0.77, and unstructured 0.53. These results demonstrate that changes in gait type affect step detection to a larger degree than changes in sensor position.

### 4.2. Steps and Shifts

The proportion of steps and shifts present in each of the three gait types is shown in Table 5. Within the experiment, each activity was designed to take approximately 10 min to complete. Because activities were standardized on time rather than number of steps, the step count for each activity varies. An average of 1050 steps were taken in the regular gait activity by each participant, 667 in the semi-regular gait activity, and 175 in the unstructured gait activity. The recorded motions in the regular gait activity were 99.6% steps. The shifts that do occur in regular gait are primarily composed of the starting and stopping of steps taken by each participant.

In the semi-regular gait activity, the number of shifts increased to 17.1% of the total steps taken. The majority of the steps recorded were taken while walking down hallways and up and down staircases, which more closely resembles regular gait. However, shifts increased because there were a larger number of starting and stopping steps taken as participants stopped to open doors throughout the activity. In addition, while hunting for the hidden items throughout rooms, participants started, stopped, shifted their weight and pivoted.

In the unstructured activity, the percentage of shifts increased again, where 35.4% of the motions recorded were shifts. In this activity, participants spent most of their time building the Lego toy at a central location, periodically interrupted with brief periods of walking as participants collected additional pieces from bins located around the room. While building, participants shifted their weight, causing some shifts. The starting and stopping steps taken while getting additional pieces were the source of many of the shifts. Another significant portion of the shifts resulted from pivots performed when participants picked up a bin of pieces and turned to walk back to the main location to continue building.

The accuracy of the pedometer algorithms evaluated changes depending on whether shifts are included or excluded as ground truth steps. Shifts are a subclass of steps that we have classified, but other works have not identified these. Some prior experiments were designed such that what we define as a shift would not appear, but other experiments have included motions that we would call shifts and treated them as steps. In our evaluation, when only steps are counted and accuracy is averaged across all metrics other than gait type, the overall RCA for regular gait is 0.97, semi-regular gait is 1.19, and unstructured gait is 3.29. These values change to 0.96 for regular gait, 0.99 for semi-regular gait, and 2.13 for unstructured gait if both steps and shifts are counted as ground truth steps. This could indicate that, in semi-regular gait, the accelerometer signal for many of the shifts generally resembles that of steps.

To demonstrate the effect shifts can have on pedometer accuracy, we evaluated a commercial pedometer (Fitbit Charge 2) by having participants wear the pedometer throughout the activities performed and comparing the reported step count to the actual step count. The Fitbit demonstrated an RCA of 0.96±0.07 (accuracy ± standard deviation per participant) for regular gait, 0.90±0.10 for semi-regular gait, and 0.65±0.15 for unstructured gait when all steps and shifts are included. When only steps are counted in the analysis, Fitbit accuracy improved to 0.97±0.07 for regular gait, 1.08±0.21 in semi-regular gait (indicating a slight overestimation of steps), and 1.00±0.38 for unstructured gait. These results suggest that the algorithm used in Fitbit devices may be designed to ignore shifts, possibly because they are difficult to detect and count, but this is a sub-optimal solution. It would be best to count them accurately, especially because they comprise 17–35% of steps in non-regular gaits.

## 5. Discussion

This work evaluates pedometer accuracy across multiple types of gait through the use of sensors located on each participant’s wrist, hip and ankle. During regular gait, we found that varying sensor position resulted in an error rate ranging from 1–9%, similar to other studies examining how this condition affects pedometer accuracy. However, when examining semi-regular or unstructured gaits common in everyday life, we saw a substantially larger error rate in pedometer accuracy. While error ranged from 3–8% during regular gait, it ranged from 7–34% during semi-regular gait and 11–466% during unstructured gait. We evaluated the same pedometer algorithms on two additional datasets that examined regular gaits and found error ranging from 6–11% (MAREA dataset) and 3–13% (Kluge dataset). This finding, combined with prior studies, indicates that pedometer error rates are similar across many metrics, including health, age, walking surface, location worn, algorithm and distance travelled, but they vary significantly when gait changes. The gait types recorded in this dataset are not available in any other public dataset, and evaluating pedometer algorithms on these gaits provides valuable feedback regarding the accuracy of pedometer algorithms designed for sensors worn throughout everyday life (including smartwatches and most fitness trackers).

Three approaches to step detection (peak detection, threshold crossing and autocorrelation) were tested on each gait type. It was found that during regular gait, each algorithm demonstrated relatively similar error rates (3–8%), but during semi-regular gait, error varied from 7–34%, and during unstructured gait, error varied from 11–466%. Specifically, the peak detection and threshold crossing algorithms both significantly overestimated steps (by as much as five times) in semi-regular or unstructured gait. The autocorrelation-based algorithm had more consistent accuracy across the semi-regular and unstructured gaits, demonstrating an error rate of less than 12%, independent of gait type. This further demonstrates the need to evaluate pedometer algorithms on different gait types, as algorithm performance can vary significantly depending on the gait being analyzed.

Training the parameters for the evaluation performed could be accomplished in multiple ways. We chose to train the parameters on the wrist-worn sensor during regular gait because it is the most common type of gait in exercise and is the type of gait that pedometer algorithms are most commonly trained for. However, regular gait is probably not optimal for everyday life because thresholds that are trained for periodicity, similarity and continuity will likely cause the algorithm to perform poorly on the more aperiodic data that is present in semi-regular and unstructured gait. One limiting factor is that the parameters for the pedometer algorithms are only trained once. This experiment demonstrates that the algorithms could benefit from switching parameter values for different types of gait. This is a topic for future work.

While the dataset presented in this work allows for more detailed evaluation of pedometer algorithms across multiple gait types and sensor positions, there are limitations. Three sensor positions—the wrist, hip and ankle—were recorded, but additional locations could be examined. In addition, the activities examined represent exercise, walking around a building, and moving within a room, but additional activities may produce additional challenges. All the types of gait examined are performed at a walking pace or slower, and the dataset only considers college-aged, healthy participants. In addition, we only simulated real-life scenarios in our experiment; we did not actually examine accuracy during everyday life.

Evaluating semi-regular and unstructured gaits is particularly challenging because it includes a sizeable portion of motions that may or may not be considered steps. We called these motions shifts to differentiate. They exist across a wide spectrum of motions that range from very step-like motions to small motions that barely resemble steps. One argument in favor of counting them is that they require energy expenditure (so contribute to exercise). An argument in favor of removing them from analysis is that they are more difficult than steps to detect and count. Either way, because they are not specifically modeled in current pedometer algorithms, we believe they contribute significantly to step count error.

Because the dataset is being made publicly available, additional algorithms can be implemented and quantitatively compared. Specifically, the identification of individual steps within our dataset allows future researchers to not only evaluate algorithms on types of gait similar to those used in free-living but to also develop and improve these algorithms through the identification of false positives and false negatives within the accelerometer signal.

Future work could include expanding the dataset to include data collected from a wider range of activities suspected to cause inaccurate step counts, such as vacuuming, mowing the lawn and folding laundry. Accelerometer data collected from sensors located in a bag, in a pocket, attached to a shoe, or in a hand could also be added. One other approach for improving accuracy may be to consider synchronizing the Shimmer3 sensors [56,57] and fusing data from all three to detect steps. Another direction for future work is identifying the differences between steps and shifts based solely on accelerometer signals. In addition, we are developing a method for actively identifying the type of gait being exhibited by a participant [58]. The detection of gait type could then be used to develop a pedometer algorithm which adjusts parameters in real time in order to provide a more accurate step count. To our knowledge, this is the first dataset and comparison of pedometer algorithms across varying types of gait representative of everyday life. The experiment performed indicates that gait type affects pedometer accuracy more than sensor position. Future work should seek to improve pedometer algorithm accuracy on the more varied gaits common in everyday life.

## Figures and Tables

**Figure 1 sensors-21-04260-f001:**
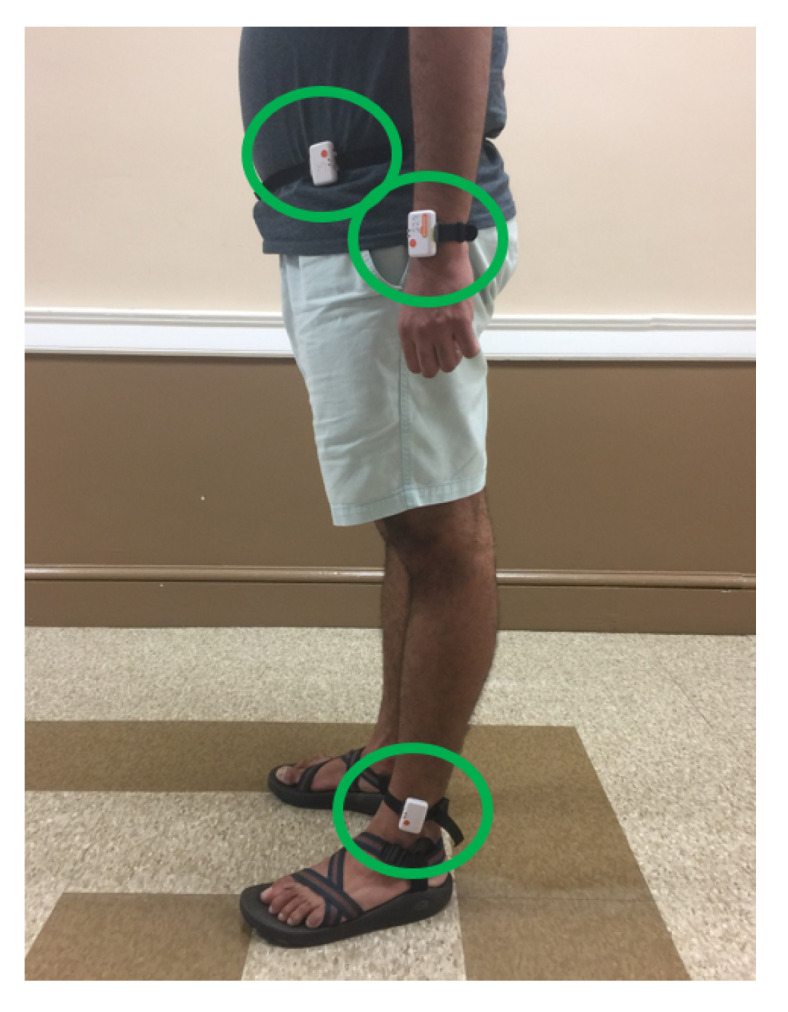
A participant wearing three Shimmer3 devices, one each on the wrist, hip and ankle.

**Figure 2 sensors-21-04260-f002:**
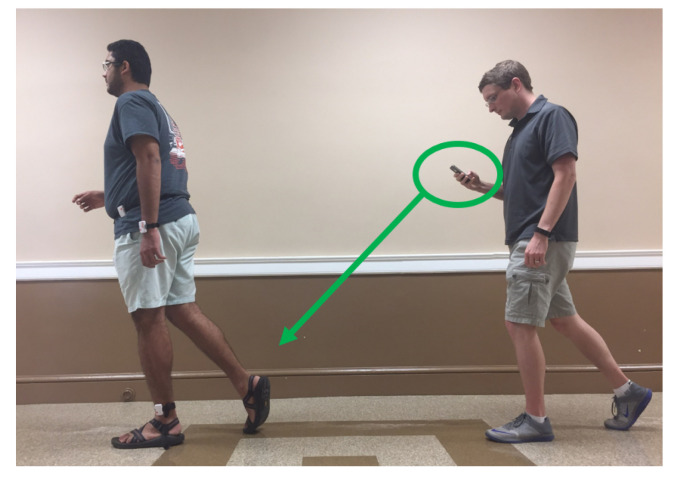
The procedure used in order to record all steps taken by a participant through each activity.

**Figure 3 sensors-21-04260-f003:**
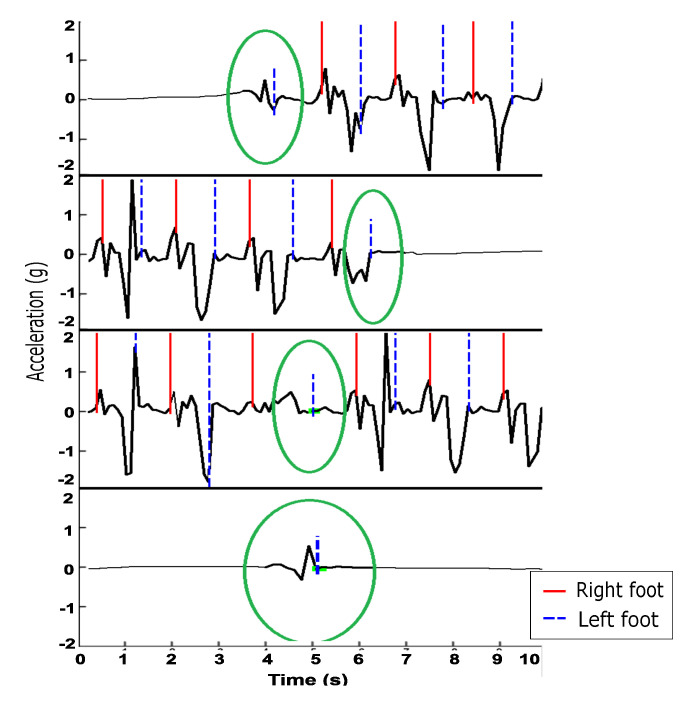
A sample from the ankle accelerometer signal associated with several ground truth steps in the regular gait activity. The peak in acceleration between each right step (red solid line) and left step (blue dashed line) is indicative of the motion of the left leg.

**Figure 4 sensors-21-04260-f004:**
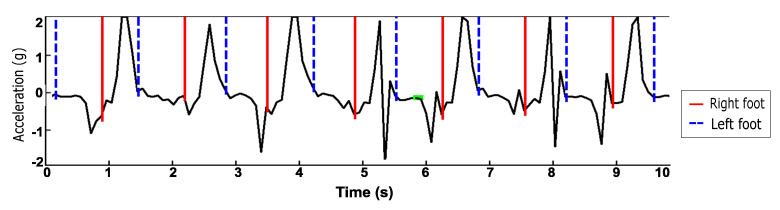
Samples taken from the ankle accelerometer signal associated with four different types of shift. The first two rows provide examples of first and last steps being taken. The third row demonstrates a pivot. The fourth row demonstrates a shuffle.

**Figure 5 sensors-21-04260-f005:**
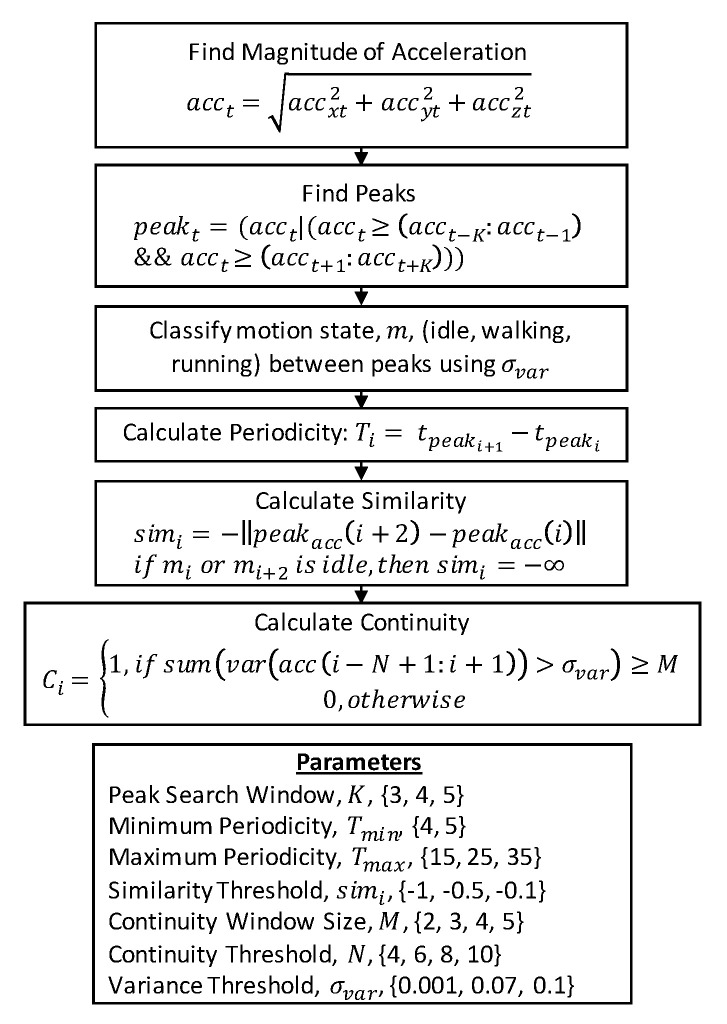
Flow chart describing the implementation of the peak detector algorithm.

**Figure 6 sensors-21-04260-f006:**
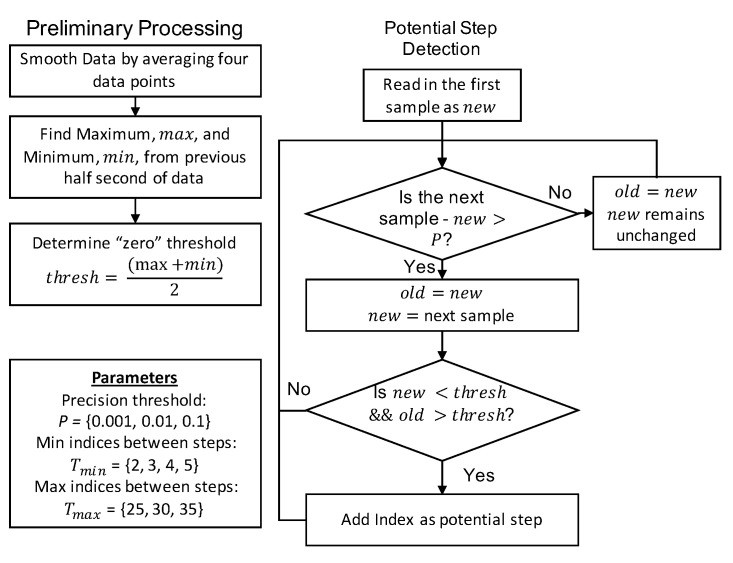
Flow chart describing the process for identifying potential steps in the threshold based algorithm.

**Figure 7 sensors-21-04260-f007:**
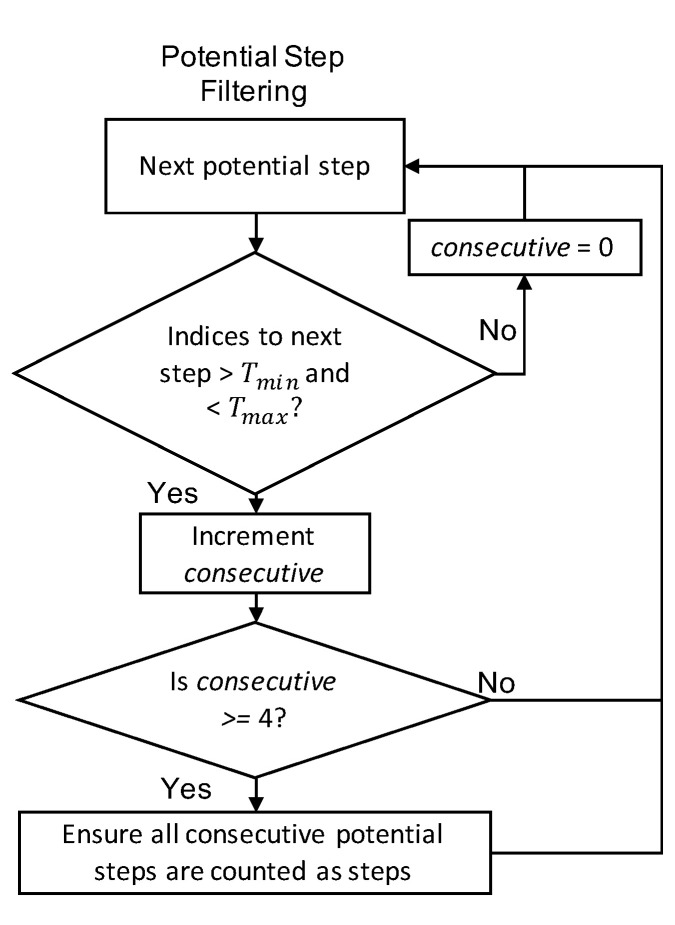
Flow chart describing the process for using time constraints to filter out potential steps for the threshold based algorithm.

**Figure 8 sensors-21-04260-f008:**
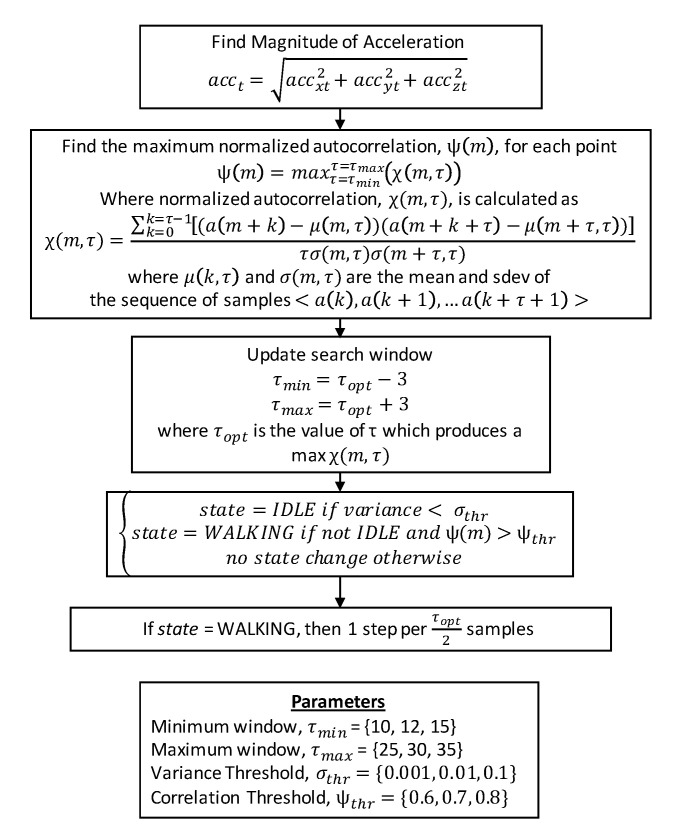
Flow chart describing the process for identifying steps in the autocorrelation algorithm.

**Figure 9 sensors-21-04260-f009:**
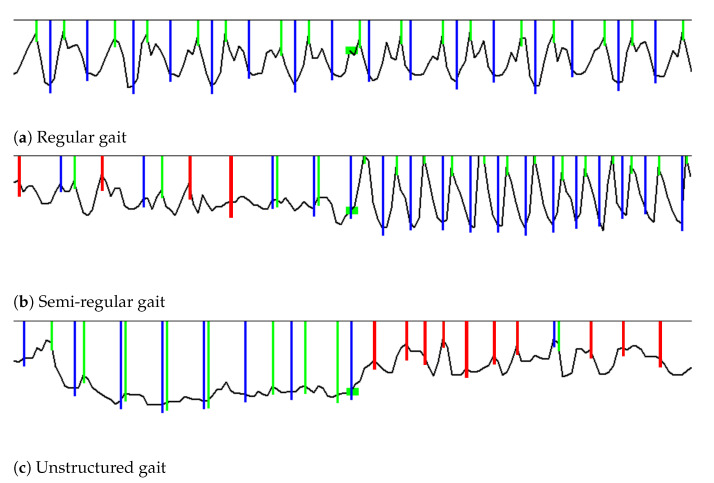
Examples demonstrating decreased accuracy during semi-regular and unstructured gait.
Blue lines represent ground truth steps. Green lines represent true positives. Red lines represent false
positives. In regular gait, all steps are detected with zero false positives, while in semi-regular and
unstructured gait there is a large number of false positives.

**Table 1 sensors-21-04260-t001:** Related works evaluating pedometer algorithm accuracy categorized by variable examined. The extent to which the variable affects accuracy is reported. Prior works have not examined regularity of gait as a variable, however this work finds that it contributes significantly to error.

Variable Examined	Citation	Error Range Found
Location worn	[22,23,24,25,26,27]	0.5–10.8%
Device model or algorithm	[23,24,28,29,30,31]	0.3–39.2%
Walking speed	[23,24,25,26,28,30,32,33,34]	0.2–96.0%
Age	[32]	3.0–19.0%
Gait aids	[35]	2.8–91.3%
Participant weight	[32]	6.6–14.7%
Type of surface	[33,34,36]	0.2–5.5%
Distance	[31]	5.4–39.18%
**Regularity of gait**	**Our work**	**5–466%**

**Table 2 sensors-21-04260-t002:** Summary of algorithms published since 2004 and algorithms selected for testing.

Type of Algorithms	Citation	Implemented
Peak detection	[15,33,34,39,40,41,42,43,44,45,46,47,48]	[15]
Threshold crossing	[16,26,27,49,50,51,52,53,54]	[16]
Autocorrelation	[17,55]	[17]

**Table 3 sensors-21-04260-t003:** Summary of criteria varied in each dataset.

Dataset	Sensor Positions	Pace	Types of Gait	Health Status
Our dataset	Wrist, hip, ankle	Walking	**Regular, Semi-reg, Unstructured**	Healthy
MAREA	Wrist, hip, ankles	**Walking, running**	Regular	Healthy
Kluge	Ankles	Walking	Regular	**Healthy, Parkinson’s**

**Table 4 sensors-21-04260-t004:** The Running Count Accuracy (accuracy ± standard deviation per participant) of three algorithm types (peak detector, threshold-crossing, and autocorrelation) across three gait types (regular, semi-regular, and unstructured) and three datasets (ours, MAREA, Kluge). Standard deviation across subjects is reported.

	Our Dataset			MAREA Dataset			Kluge Dataset		
**Algorithm**	**Regular**	**Semi**	**Unstrct**	**Regular**	**Semi**	**Unstrct**	**Regular**	**Semi**	**Unstrct**
Peak	0.92±0.11	1.30±0.21	3.09±2.55	0.94±0.07	N/A	N/A	1.03±0.35	N/A	N/A
Threshold	1.03±0.17	1.34±0.17	5.66±1.88	1.11±0.46	N/A	N/A	1.13±0.34	N/A	N/A
Autocor.	0.95±0.24	0.93±0.17	1.11±0.44	0.92±0.13	N/A	N/A	0.97±0.55	N/A	N/A

**Table 5 sensors-21-04260-t005:** Total steps and shifts manually recorded through each of the three activities across all 30 participants.

Activity	Steps	Shifts
Regular	31,401 (99.6%)	133 (0.4%)
Semi-regular	18,444 (82.9%)	3795 (17.1%)
Unstructured	4542 (64.6%)	2490 (35.4%)
Overall	54,387 (89.4%)	6418 (10.6%)

## Data Availability

Data publicly available at https://sites.google.com/view/rmattfeld/pedometer-dataset, accessed on 21 June 2021.

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
