# Peer review of "Evaluating Pedometer Algorithms on Semi-Regular and Unstructured Gaits"

_sensors, 2021, doi:10.3390/s21134260_

Round 1

Reviewer 1 Report

The authors proposed an evaluation study of pedometer algorithms. The idea sounds interesting, however, some important points are described, such as:

a) Abstract section lacks to present the paper's contribution and briefly describe the main achievements. 

b) Introduction describes the motivation and gives a little presentation about the envisaged problem. It was not possible to detect significant contributions when considered literature reviews—the strategy used by the author did not create innovations in the field. 
c) The last paragraph describing the organization of the paper is missing in the Introduction section. 
d) Related works are missing.
e) Section 2 is only a setup configuration, and appropriately formal definitions are missing. 
f) It is not possible to detect where results come from. This occurs because the proposal was not formally described. 
g) Conclusion is not supported by the data.
h) The paper has several flaws. The organization and presentation of the proposal are chaotic, and a logic sequence was not defined. A scientific methodology in order to present the proposal and results is missing too. 
i) References are outdated. 

Author Response

The authors proposed an evaluation study of pedometer algorithms. The idea sounds interesting, however, some important points are described, such as:

  1. a) Abstract section lacks to present the paper's contribution and briefly describe the main achievements. 

We have updated the abstract (lines 15-17) to clarify our contribution:

“The main contribution of this paper is to evaluate pedometer algorithms when the consistency of gait changes to simulate everyday life activities other than exercise. In our study, we found that semi-regular and unstructured gaits resulted in 5-466\% error.”

  1. b) Introduction describes the motivation and gives a little presentation about the envisaged problem. It was not possible to detect significant contributions when considered literature reviews—the strategy used by the author did not create innovations in the field. 

Our contributions have been clarified with the following two places in the introduction:

1) Lines 38-43:

“The main contribution of this paper is that we are the first to look at the effect of regularity of gait on pedometer accuracy. All previous works evaluate accuracy while participants walk in a regular gait resembling mild exercise. Specifically, subjects were asked to walk for a period of time or distance or step count, with no breaks or other intermittent activities. We hypothesize that disruptions in walking throughout everyday life are a large contributor to pedometer inaccuracy.”

2) Lines 71-73:

“The main contribution of this paper is to evaluate pedometer algorithms when the consistency of gait changes to simulate everyday life activities other than exercise.”

  1. c) The last paragraph describing the organization of the paper is missing in the Introduction section. 

We have added a paragraph describing the organization of the paper at the end of the introduction (lines 78-84) as follows:

“This paper is organized as follows. In Section 2, we provide related works and identify regularity of gait as a variable in pedometer algorithm evaluation that has not been researched. In section 3, we describe the data collection process, ground truth step identification process, and the process used to evaluate 3 state of the art pedometer algorithms. Section 4 provides the results of evaluating the 3 pedometer algorithms on our dataset as well as two other publicly available datasets. Section 5 discusses our findings.”

  1. d) Related works are missing.

We have created a related works section after the introduction.

  1. e) Section 2 is only a setup configuration, and appropriately formal definitions are missing. 

We define regular gait as what occurs during exercise and other daily activities having extended periods (5+ min) of consistent walking that is uninterrupted. We define semi-regular gait as what occurs when moving through buildings, which is primarily composed of periods of time that resemble regular gait broken up by brief interruptions (e.g. stopping, starting, or changing direction). We define unstructured gait as what occurs when performing activities within a room, which is primarily composed of very brief periods of regular gait (approximately 3-10 steps) with more frequent interruptions including periods of rest and change of direction.

  1. f) It is not possible to detect where results come from. This occurs because the proposal was not formally described. 

The “Metrics” subsection (2.6) is now titled “Evaluation”. In addition, the following sentence was added (lines 266-268) for clarity:

“Each of the pedometer algorithms were evaluated by comparing the number of steps detected by the algorithm against the number of steps actually taken according to the ground truth labels for each participant and activity.”

  1. g) Conclusion is not supported by the data.

Our main conclusion is that semi-regular and unstructured gait are a major cause of pedometer inaccuracy.  Our data show 3-8% error during regular gait, rising to 7-34% during semi-regular gait and 11-466% during unstructured gait.  We believe this data supports our conclusion.

  1. h) The paper has several flaws. The organization and presentation of the proposal are chaotic, and a logic sequence was not defined. A scientific methodology in order to present the proposal and results is missing too. 

We have added a paragraph describing the organization of the paper at the end of the introduction (lines 78-84) as described above.

  1. i) References are outdated. 

20% of our references are from 2019 or more recent, 51% are from 2015 or more recent, and 78% are from 2010 or more recent.

Reviewer 2 Report

This work evaluates pedometer accuracy across multiple types of gait through the use of sensors located on each participant’s wrist, hip, and ankle. The whole paper is well organized, and the contribution is interesting. However, the experiment is limited. I vote for major revision and the following suggestions need to be considered.

1. Should add more comparison results with atate-of-the-art methods.
2. Please highlight the contributions of this paper in Introduction.
3. Separate a 'related work' section from introduction.
4. Please list run time of experiment.
5. Please list some failure examples and explain why.

Author Response

This work evaluates pedometer accuracy across multiple types of gait through the use of sensors located on each participant’s wrist, hip, and ankle. The whole paper is well organized, and the contribution is interesting. However, the experiment is limited. I vote for major revision and the following suggestions need to be considered.

  1. Should add more comparison results with state-of-the-art methods.

Comparing results of pedometer algorithms is one of the key challenges in this field that we are hoping to help change by making our dataset public. Comparing results across papers is frequently misleading because the data collected can vary significantly and authors very rarely make their datasets public to allow for accurate, direct comparisons.

We have added this sentence in the introduction (Line 73):

“Finally, we recognize that many results from wearable device analyses are inhomogeneous and in an effort to provide opportunities for algorithms to be tested on identical datasets, we have made our data publicly available at https://sites.google.com/view/rmattfeld/pedometer-dataset.”

Please see: https://doi.org/10.1016/j.measurement.2020.107789 for a thorough discussion of this problem.

  1. Please highlight the contributions of this paper in Introduction.

1) Lines 38-43 have been modified to clarify the contribution of this paper.

“The main contribution of this paper is that we are the first to look at the effect of regularity of gait on pedometer accuracy. All previous works evaluate accuracy while participants walk in a regular gait resembling mild exercise. Specifically, subjects were asked to walk for a period of time or distance or step count, with no breaks or other intermittent activities. We hypothesize that disruptions in walking throughout everyday life are a large contributor to pedometer inaccuracy.”

2) Lines 71-73 have been added to provide further clarity.

“The main contribution of this paper is to evaluate pedometer algorithms when the consistency of gait changes to simulate everyday life activities other than exercise.”

  1. Separate a 'related work' section from introduction.

We have created a related works section after the introduction.

  1. Please list run time of experiment.

It takes less than 1 minute to run a step detection algorithm on all 90 trials (30 participants with 3 variations on the regularity of gait each). Variations in CPU hardware will affect specific timings.

  1. Please list some failure examples and explain why.

We added an additional figure and paragraph in section 4.1 to provide some failure examples. Please see Figure 9 in the updated document as well as the following paragraph from lines 313-323:

“Figure 9 demonstrates some true positive and false positive examples for the peak detector algorithm on regular gait, semi-regular gait, and unstructured gait. Blue lines indicate ground truth steps. Green lines indicate true positive steps detected by the peak detector algorithm, and red lines indicate false positives. In 9a, it can be seen that for regular gait, the detected steps are all true positives. In 9b, it can be seen that during semi-regular gait, as the participant opened a door at the end of a hall, transitioning from walking down a hall to walking on stairs, four false positives were detected. In 9c, it can be seen that while the participant was walking during unstructured gait, the peak detector did well, but it picked up many false positives as the participant constructed their Lego. The false positives were caused by detecting peaks in acceleration when no steps were taken.”

Reviewer 3 Report

The paper contains a valuable contribution. The subject is within the scope of the journal and the objective of research is well stated. However, some clarifications about the underlying hypothesis / scope are needed.

In the opinion of this Reviewer the manuscript deserves to be published once the Author takes into account the raised issues. Some important details are missing.

Abstract:

  1. Row 17. Check the values reported (5-566%). They are not consistent with the values reported in the discussion section.

Introduction / Literature review

  1. The research scope is clear as well as the literature review. Anyway, the authors should better highlight the innovative aspects of their work in the manuscript.

What are the advantages / improvements in the proposed approach, which are not covered by the current studies?

  1. For the sake of readability, at the end of Section 1 the authors should describe how the paper is structured.

Methods

  1. No details on how the data are sampled from the Shimmer3 devices (sampling rate, filtering, full scale range, data concentrator, data fusion, accuracy, etc.) are given.
  2. Another important aspect to be clarified is the synchronization among the Shimmer3 motes themselves and the Fitbit Charge 2 watch.

Synchronization is an important issue in such a system and a lot of papers analyze it in detail (e.g. https://doi.org/10.1109/MELECON48756.2020.9140622, document that could be cited in the text).

How much does synchronization problem could affect the entire system?

  1. Could the proposed solution gain better results with a multi-unit synchronized system for activity monitoring (e.g., https://doi.org/10.1109/JSEN.2020.2982744, document that could be cited in the text)?
  2. Figure 3-4. Use the lowercase g for indicating the gravity acceleration.
  3. Figure 3-4. Add a legend.

Minor

  1. Mainly the English is good and there are only a few typos. However, the paper should be carefully rechecked.

Author Response

The paper contains a valuable contribution. The subject is within the scope of the journal and the objective of research is well stated. However, some clarifications about the underlying hypothesis / scope are needed.

In the opinion of this Reviewer the manuscript deserves to be published once the Author takes into account the raised issues. Some important details are missing.

Abstract:

  1. Row 17. Check the values reported (5-566%). They are not consistent with the values reported in the discussion section.

The values reported in the abstract have been corrected to match those in the discussion. Thank you for pointing this out.

Introduction / Literature review

2. The research scope is clear as well as the literature review. Anyway, the authors should better highlight the innovative aspects of their work in the manuscript.

What are the advantages / improvements in the proposed approach, which are not covered by the current studies?

The current body of work related to step detection algorithms entirely omits evaluating the algorithms while changing the consistency of gait. However, most pedometer algorithms are used throughout the day, and everyday motions include frequent interruptions to gait. Pedometer algorithms need to be evaluated in these conditions to find their accuracy as they are actually used in daily life. We created a dataset and demonstrated that when gait is less consistent, the current state-of-the-art algorithms have much more significant error rates than they do when evaluated across the “traditional” error sources investigated by prior studies (ex: location worn, device model, walking speed, age, health, weight, type of surface, and distance). We have modified the introduction to make this clearer in two places:

1) Lines 38-43 have been modified to clarify the contribution of this paper.

“The main contribution of this paper is that we are the first to look at the effect of regularity of gait on pedometer accuracy. All previous works evaluate accuracy while participants walk in a regular gait resembling mild exercise. Specifically, subjects were asked to walk for a period of time or distance or step count, with no breaks or other intermittent activities. We hypothesize that disruptions in walking throughout everyday life are a large contributor to pedometer inaccuracy.”

2) Lines 71-73 have been added to provide further clarity.

“The main contribution of this paper is to evaluate pedometer algorithms when the consistency of gait changes to simulate everyday life activities other than exercise.”

3. For the sake of readability, at the end of Section 1 the authors should describe how the paper is structured.

We have added a paragraph describing the organization of the paper at the end of the introduction (lines 78-84) as follows:

“This paper is organized as follows. In Section 2, we provide related works and identify regularity of gait as a variable in pedometer algorithm evaluation that has not been researched. In section 3, we describe the data collection process, ground truth step identification process, and the process used to evaluate 3 state of the art pedometer algorithms. Section 4 provides the results of evaluating the 3 pedometer algorithms on our dataset as well as two other publicly available datasets. Section 5 discusses our findings.”

Methods

4. No details on how the data are sampled from the Shimmer3 devices (sampling rate, filtering, full scale range, data concentrator, data fusion, accuracy, etc.) are given.

5. Another important aspect to be clarified is the synchronization among the Shimmer3 motes themselves and the Fitbit Charge 2 watch.

Points 4 and 5 were addressed with the following addition to the data collection section (lines 126-129):

“The Shimmer3 devices recorded at 15Hz with raw acceleration measurements ranging from -2 to 2 Gravities and a noise density of 125 µg/Hz. They were synchronized to a single computer's clock.”

The Fitbit Charge 2 watch was not synchronized with the Shimmer3 devices because the raw Fitbit data was not available. The steps reported by the Fitbit were simply recorded before and after the activity performed. The Shimmer3 devices were synchronized with a computer’s clock. This information has been added to the methods section.

It should be noted that we aim for the sensors to be evaluated separately. Users of commercial pedometers do not typically wear multiple devices simultaneously, and we are not proposing a system that requires the user to wear multiple sensors and fuse their data. While this may achieve better results, it is not the purpose of this experiment. However, this consideration has been added to the discussion section as a potential future work (lines 443 – 444).

Synchronization is an important issue in such a system and a lot of papers analyze it in detail (e.g. https://doi.org/10.1109/MELECON48756.2020.9140622, document that could be cited in the text).

How much does synchronization problem could affect the entire system?

6. Could the proposed solution gain better results with a multi-unit synchronized system for activity monitoring (e.g., https://doi.org/10.1109/JSEN.2020.2982744, document that could be cited in the text)?

It is possible using a multi-unit synchronized system could improve results. However, this was not the purpose of this study and each sensor’s data was considered independently within this work. This has been added to the discussion section (lines 443-444) as a potential future approach, including the citations requested:

“One other approach for improving accuracy may be to consider synchronizing the Shimmer3 sensors and fusing data from all 3 to detect steps.”

7. Figure 3-4. Use the lowercase g for indicating the gravity acceleration.

8. Figure 3-4. Add a legend.

Figures 3-4 have been updated with a lowercase g to indicate gravity acceleration and a legend has been added.

Minor

9. Mainly the English is good and there are only a few typos. However, the paper should be carefully rechecked.

The paper has been reviewed for typos and some small fixes have been made.

Round 2

Reviewer 1 Report

The authors implemented all suggestions indicated in the first revision round, thus, it is possible to note the overall quality of the paper was improved. However, the comparison between literature solutions and the paper is fuzzy and the innovations are only borderlines when considered literature review. It is strongly recommended the authors add a Table (in section 2) comparing in an explicit way all literature contributions about the topic and highlight the innovation of the proposal.  

Additionally, a subsection about the experimentation setup and validity threats. The former is essential for reproducibility issues of results whereas the latter is critical to indicate the bottleneck and limitation of the proposal. 

Author Response

The authors implemented all suggestions indicated in the first revision round, thus, it is possible to note the overall quality of the paper was improved.

However, the comparison between literature solutions and the paper is fuzzy and the innovations are only borderlines when considered literature review. It is strongly recommended the authors add a Table (in section 2) comparing in an explicit way all literature contributions about the topic and highlight the innovation of the proposal.  

We have added a new table in the related works section (Table 1) and the following sentences in lines 88-92 describing the information shown:

“These variables have all been found to influence pedometer accuracy as seen in Table 1. In this table it can also be seen that our work is the first to examine pedometer accuracy as regularity of gait changes. It can also be seen that the regularity of gait affects pedometer accuracy to a larger degree than previously examined variables.”

Additionally, a subsection about the experimentation setup and validity threats. The former is essential for reproducibility issues of results whereas the latter is critical to indicate the bottleneck and limitation of the proposal. 

We have added a subsection at the start of our Methods section (lines 118-128) describing the experimentation process:

“The goal of this experiment is to evaluate the effect of a novel source of error, regularity of gait, on standard pedometer algorithms. First, we instrument participants with accelerometers and have them conduct activities that vary the regularity of gait. Second, we use synchronized video to annotate the accelerometer data denoting the ground truth times of all steps. Third, we search a publication database to identify a representative set of popular pedometer algorithms.  Lastly, we evaluate the accuracy of these pedometer algorithms as the error source is varied.  Note that the goal of this experiment is to determine the effect of the error source on pedometer accuracy.  It is not our intent to identify the best possible pedometer algorithm.  We seek to determine if the novel error source is something that needs to be considered in future pedometer algorithm designs.”

We have also added a sentence to our limitations paragraph in the discussion section as follows (Lines 436-438):

“In addition, we only simulated real-life scenarios in our experiment; we did not actually examine accuracy during everyday life.”

Reviewer 2 Report

Still need more comparison results with state-of-the-art methods. The authors can implement open-source code in their own dataset. Otherwise, it is hard to demonstrate the performance and I will consider to reject this.

Author Response

Still need more comparison results with state-of-the-art methods. The authors can implement open-source code in their own dataset. Otherwise, it is hard to demonstrate the performance and I will consider to reject this.

Thank you for your feedback. The three algorithms we implemented are state-of-the-art, and they are representative of the common approaches used in step detection algorithms. We are not intending to test all of the latest pedometer algorithms – instead, we aim to identify if regularity of gait is a variable that can impact pedometer accuracy. We have added lines 124-128 to clarify this intent.

With time as a constraint (we only have 5 days to make all replies to reviewers), implementing additional algorithms is not feasible (units must be adjusted, sample rate must be adjusted for, the format in which the data is read must be adjusted, verification of results, etc).

We believe the finding of this paper – which is that the regularity of gait can significantly impact pedometer accuracy – is valid based on usage of the three state-of-the-art algorithms used. These three algorithms utilize the most common approaches to step detection. We believe our finding identifies a critical gap in pedometer algorithms that needs to be addressed in future research.

Reviewer 3 Report

Authors have strongly improved the article according with the reviewers' comments.

Author Response

Authors have strongly improved the article according with the reviewers' comments.

Thank you for your valuable feedback that has helped us improve the paper.

Round 3

Reviewer 1 Report

The reviewer suggests accepting the paper in the present form.

Reviewer 2 Report

The revised version can be acceptable.